# Alu–Mediated Duplication and Deletion of Exon 11 Are Frequent Mechanisms of *PALB2* Inactivation, Predisposing Individuals to Hereditary Breast–Ovarian Cancer Syndrome

**DOI:** 10.3390/cancers16234022

**Published:** 2024-11-30

**Authors:** Diletta Sidoti, Valeria Margotta, Diletta Calosci, Erika Fiorentini, Costanza Bacci, Francesca Gensini, Laura Papi, Marco Montini

**Affiliations:** 1Department of Biomedical, Experimental and Clinical Sciences “Mario Serio”, University of Florence, 50139 Florence, Italy; diletta.sidoti@unifi.it (D.S.); valeria.margotta@unifi.it (V.M.); diletta.calosci@unifi.it (D.C.); erika.fiorenini@unifi.it (E.F.); francesca.gensini@unifi.it (F.G.); marco.montini@unifi.it (M.M.); 2Medical Genetics Unit, SOS Day-Service Palagi, USL Toscana Centro, 50122 Florence, Italy; costanza.bacci@uslcentro.toscana.it

**Keywords:** *PALB2*, Alu repeats, large genomic rearrangements, breast cancer

## Abstract

Large genomic rearrangements of the *PALB2* gene, especially deletions and duplications, are associated with hereditary breast–ovarian cancer. This study investigates intronic breakpoints linked to rearrangements in *PALB2* exon 11, which is crucial for understanding the mechanisms affecting patients with hereditary breast and ovarian syndrome. Through next-generation sequencing, one duplication and three deletions were identified, confirmed by Multiplex Ligation-Dependent Probe Amplification. Detailed characterization revealed a tandem duplication of 5134 base pairs mediated by *AluY* repeats and identical deletions in three unrelated patients promoted by *AluSx* elements, resulting in a truncated *PALB2* protein. These findings highlight the instability of intronic regions flanking exon 11 and suggest directions for future research on the prevalence and functional implications of these genomic alterations.

## 1. Introduction

Hereditary breast and ovarian cancer syndrome (HBOC) increases susceptibility to multiple malignancies in both women and men primarily involving breast and ovarian cancers but also prostate and pancreatic cancers [1,2]. The key features of HBOC syndrome are early disease onset (often before age 36), bilateral presentations, and a familial pattern with multiple affected individuals [3]. HBOC is inherited in an autosomal dominant manner and is frequently linked to pathogenic variants (PVs) in the *BRCA1* and *BRCA2* genes, essential for repairing double-stranded DNA breaks via homologous recombination (HR) [4,5,6,7].

Recently, *PALB2* has been recognized as a third highly penetrant gene [8]. *PALB2* plays a critical role in the HR pathway by serving as a scaffold within a large molecular complex that includes both *BRCA1* and *BRCA2* [9,10]. It also facilitates the transport of *RAD51* to chromosomal lesions to initiate repair and to form the D-loop (displacement loop), which is essential for meiotic recombination repair [11,12]. The *BRCA2*-*PALB2* dimer is essential for repair processes, including HR, DNA double-strand break repair, and the regulation of the S-phase DNA damage checkpoint. In addition to its interaction with *BRCA2* through the WD40 domain at the C-terminus, *PALB2* also binds to *BRCA1* via a coiled coil motif at the N-terminus. Disruption of this complex can significantly reduce the efficiency of the HR repair system [8,12,13,14].

Germline biallelic PVs in *PALB2* lead to the rare autosomal recessive form of Fanconi Anemia, which is characterized by genomic instability, early bone marrow failure, growth issues, and an increased risk of cancer [15,16,17]. Instead, monoallelic *PALB2* PVs can be found in 0.4% to 3.9% of breast cancer patients and have been linked to a 41–60% risk of breast cancer, a 3–5% risk of ovarian cancer, and 2–5% risk of pancreatic cancer [1,3,7,18]. Carriers of germline PVs in *PALB2* often exhibit aggressive clinical and pathological features, including a triple-negative breast cancer phenotype and an earlier age of diagnosis [9,19].

The National Comprehensive Cancer Network (NCCN) guidelines recommend that women with *PALB2* PVs begin annual mammography and MRI with contrast starting at age 30. Risk-reducing mastectomy may be discussed with the patient, and, although data on risk-reducing salpingo-oophorectomy are limited, it may be considered starting at 45 to 50 years of age [1].

In *PALB2* mutational screening, the molecular characterization of PVs has mainly focused on single nucleotide substitutions and small insertions and deletions. However, a smaller proportion of large genomic rearrangements (LGRs), i.e., deletions and duplications, can also significantly influence the mutational landscape [20,21,22]. A few studies reported *PALB2* germline LGRs in patients with breast, pancreatic, and ovarian cancers [10,21,23,24,25,26,27,28,29,30,31,32,33]. Yang et al. [32] identified 23 germline *PALB2* LGRs in an international study of 524 families with pathogenic *PALB2* variants: all of them were clustered in the WD40 domain and five of them involved only exon 11 (specifically, four deletions and one duplication). Several other studies confirmed that *PALB2* rearrangements occur mainly in the sequences between exon 7 and exon 13 which encode the WD40 domain, and rearrangements selectively involving exon 11 were reported in an additional five cases, specifically four duplications and one deletion [10,21,23,24,25,26,27,28,29,30,31,32,33].

We routinely perform genetic screening of *PALB2* in patients who test negative for PVs in *BRCA1* and *BRCA2*. Our analysis identified three women carrying a deletion involving exclusively *PALB2* exon 11, and one with a duplication of the same exon.

Given the frequent identification of exon 11 rearrangements in *PALB2*, we aimed to precisely characterize the intronic breakpoints of the rearrangements to elucidate the molecular mechanisms driving these genomic alterations. Additionally, we conducted transcript-level assessments through RNA analysis, providing valuable insights into potential splicing variations and allelic expression patterns.

## 2. Materials and Methods

### 2.1. Patients

Four women, evaluated due to their medical history of cancer(s), were involved in this study. Since they tested negative for PVs in *BRCA1* and *BRCA2*, we investigated a panel of 12 additional genes known to give a high to moderate risk for HBOC. Pedigree of the probands is given in Figure 1.

Patient 1 was diagnosed with breast cancer at 36 years old, with a contralateral involvement at 41. At 38 years old, she developed thyroid cancer, and ten years later, she manifested colorectal cancer. She had gastric cancer at 59 years of age and a third breast cancer at 62. Moreover, she was diagnosed at 62 and 63 with two primary lung cancers and is currently undergoing treatment. Three of her sisters died from various HBOC- and colon-related malignancies between the ages of 54 and 61. Her father, paternal uncles, and cousins also had a history of breast, pancreatic, and prostate cancers (Figure 1a). The patient tested negative for PVs in genes related to Lynch Syndrome and *MUTYH*-associated polyposis.

Patient 2 presented with a history of breast cancer at 41 years old, with a relapse at 48 years old. At 56 years old, she was diagnosed with pancreatic neoplasia. Her father passed away at 77 years old due to pancreatic cancer diagnosed 3 years earlier, and her father’s aunt developed breast cancer at the age of 87 and died two years later (Figure 1b).

Patient 3 was diagnosed with breast cancer and pancreatic cancer at 51 and 60 years old, respectively. In her family, her sister had breast cancer at 50 years old and one paternal cousin developed breast cancer at the age of 70 (Figure 1c).

Patient 4 had their first breast cancer diagnosis at the age of 38 and contralateral breast cancer at 43 years old. Her mother was diagnosed with breast cancer at 38 years old and she passed away four years later (Figure 1d).

### 2.2. DNA Extraction and NGS Analysis

The patients’ samples were collected in tubes containing EDTA. Genomic DNA (gDNA) was extracted from peripheral blood samples using the QIAsymphony SP/AS nucleic acid extraction tool (Qiagen, Hilden, Germany) in accordance with the manufacturer’s instructions. Next-generation sequencing was performed using the Devyser BRCA and HBOC kits (Devyser AB, Stockholm, Sweden) with target enrichment probes for *BRCA1/BRCA2* and the following 12 genes: *ATM*, *BARD1*, *BRIP1*, *CDH1*, *CHEK2*, *NBN*, *PALB2*, *PTEN, RAD51C*, *RAD51D*, *STK11*, and *TP53*. Gene-targeted enrichment was performed following the manufacturer’s guidelines. Massively parallel sequencing was carried out using the MiSeq2000 instrument (Illumina, San Diego, CA, USA), achieving an average read depth of 500X. FASTQ outcomes analysis was performed using CE-IVD-certified Amplicon Suite bioinformatic analysis software, v3.7.0 (SmartSeq srl, Novara, Italy).

### 2.3. Multiplex Ligation-Dependent Probe Amplification (MLPA)

The verification of predicted LGRs was carried out on genomic DNA (gDNA) patient’s samples using the SALSA MLPA P260 PALB2-RAD50-RAD51C-RAD51D Probe mix (MRC Holland, Amsterdam, The Netherlands), according to the manufacturer’s guidelines. The amplicons were analyzed on SeqStudio genetic analyzer, v1.2.4 (Applied Biosystem, Thermo Fisher, Monza, Italy), and data were processed using Coffalyser.Net software, v220513.1739 (MRC Holland, Amsterdam, The Netherlands).

### 2.4. RNA Workflow

RNA was isolated from peripheral blood using PAXgene^®^ blood RNA tubes (BD Biosciences, Meylan, France) and extracted using PAXgene^®^ blood RNA Kit Pre Analytix (Qiagen, Hilden, Germany) with DNase treatment as per the manufacturer’s instructions. Extracted RNA samples were quantified using the NanoDrop^®^ ND-1000 (NanoDrop^®^ Technologies) spectrophotometer, v3.8 (Thermo Fisher Scientific, Wilmington, DE, USA), followed by a confirmatory quantification with the Qubit High Sensitivity (HS) Assay Kit (Thermo Fisher Scientific, Waltham, MA, USA).

Complementary DNA (cDNA) was synthesized using the SuperScript™ VILO™ cDNA Synthesis Kit from Invitrogen™ (Thermo Fisher Scientific, Waltham, MA, USA) containing a genetically modified MMLV SuperScript™ III reverse transcriptase to improve its performance. The cDNA was then amplified using primers designed to target exon 11 deletions and duplication (forward on exon 9: 5′ TGGGACCCTTTCTGATCAAC 3′; reverse on exon 12: 5′ TGCCCTGGAGGAAGACAGTA 3′). PCR amplification was performed using the Hot Start Taq DNA Polymerase kit (Qiagen, Hilden, Germany). PCR products were verified on 2% agarose gel, visualized by ethidium bromide staining, and sequenced by Sanger sequencing.

### 2.5. Sequencing Analysis of Gene Deletion Breakpoints

To amplify the expected breakpoints of the gene rearrangements identified by NGS and MLPA analyses, forward and reverse primer pairs were designed, respectively, on the intronic 10 and 11 regions lacking *Alu* sequences, which were masked using the online RepeatMasker tool (URL (accessed on 15 February 2024): https://www.repeatmasker.org/). In Appendix A, we show the schematic representation of the primer walking strategy used to identify the deletion intronic breakpoints: six forward primers on intron 10 (from 10R to 10Z) and six reverse primers on intron 11 (from 11F to 11A). Appendix A gives the primer sequences.

Long-range PCRs were performed using the Q5^®^ High-Fidelity DNA Polymerase (New England Biolabs, Ipswich, MA, USA), and PCR products were sequenced by Sanger sequencing.

Briefly, PCR products were sequenced on the SeqStudio Genetic Analyzer sequencer (Applied Biosystems, Thermo Fisher, Monza, Italy) using BigDye Terminator Chemistry (Applied Biosystems, Thermo Fisher Scientific, Monza, Italy) according to the manufacturers’ recommendations. The results were analyzed with the SeqScanner 2 v2.0 and SeqScape v2.6 software (Applied Biosystems, Thermo Fisher Scientific, Monza, Italy).

### 2.6. Microsatellite Analysis and Haplotyping

Microsatellites on chromosome 16, flanking *PALB2*, were investigated using microsatellites D16S3075, D16S3103, D16S3046, D16S3068, D16S3136, and D16S415 from the ABI PRISM Linkage Mapping set v2.5 (Life Technologies, Carlsbad, CA, USA) (Table 1). The amplification products were separated by capillary electrophoresis on a SeqStudio Genetic Analyzer, v1.2.4 (Applied Biosystems, Waltham, MA, USA) and analyzed using the Gene Mapper software v6.0 (Applied Biosystems, Waltham, MA, USA).

A total of seven individual carriers of the *PALB2* exon 11 deletion were genotyped. Haplotypes were constructed manually from microsatellite analyses, assuming the least number of possible recombinations.

### 2.7. Interpretation of Variants

Mutation nomenclature follows the Human Genome Variation Society (URL (accessed on 2 October 2023): https://hgvs-nomenclature.org/stable/) recommendations [34]. The DNA mutation numbering is based on the *PALB2* cDNA sequences (NM_024675.3) with the A of the ATG translation-initiation codon numbered as +1. Amino acid numbering starts with the translation initiator methionine as +1.

## 3. Results

### 3.1. NGS Analysis and MLPA LGRs Confirmation

Four women were referred to our laboratory due to their medical history, which was noticeable by several HBOC-related cancers also affecting other family members. Initial genetic screening for BRCA1 and BRCA2 did not reveal any mutations in these genes. Thus, to identify the potential presence of PVs in additional syndrome-associated genes that could explain the clinical phenotype, we analyzed the patient’s DNAs using the Devyser HBOC genes panel assay. The results of the NGS analysis revealed large rearrangements involving exon 11 of the PALB2 gene in all four patients. Patient 1 exhibited a duplication of PALB2 exon 11 (Figure 2a), while the other three had a deletion of PALB2 exon 11 (Figure 2b). To validate the identified rearrangements, we performed MLPA that confirmed their presence (Figure 2c,d).

### 3.2. Transcript Analysis

To evaluate the consequences of the identified LGRs at the transcriptomic level, we designed primers that specifically targeted the region of interest in the cDNA. The forward primer was positioned approximately 30 base pairs from the 3′-end of exon 9, while the reverse primer was situated in the middle of exon 12. A distinct 486 bp PCR amplification was observed in the sample with duplication, whereas 310 bp PCR amplification was evident in the probands’ samples with deletion (Figure 3a). A 398 bp PCR amplification product was detected in the control samples, which matched the wild-type allele PCR amplifications of the probands (Figure 3a).

Sanger sequencing of the duplicated cDNA amplification product confirmed that the duplication was in direct orientation and out-frame. Therefore, the variant could be described as follows: c.(3113+1_3114-1)_(3201+1_3202-1)dup, r.3114_3201dup, p.(Gly1068Glufs*14) (Figure 3b).

Sequencing of the deleted cDNA samples also revealed an aberrant transcript product showing the complete deletion of exon 11 and resulting in a frameshift: c.(3113+1_3114-1)_(3201+1_3202-1)del, r.3114_3201del, p.(Asn1039Glyfs*7) (Figure 3c).

Furthermore, we applied an algorithm to predict the possible nonsense-mediated decay (NMD) effect caused by the formation of a premature termination codon (PTC) in the mutated transcripts. For the prediction of the NMD effect, we used Masonmd (Make Sense of NMD), an R package for predicting NMD-elicit mutations. Predictive analysis confirmed the mechanism of NMD for both aberrant transcripts.

### 3.3. Breakpoints Identification of Exon 11 LGRs

To comprehend the molecular mechanism underlying the rearrangements identified in our patients, we employed two distinct strategies to accurately determine the intronic breakpoints in our deletion and duplication cases, respectively.

Regarding the duplicated case, we found the same breakpoints described by Bouras et al. [28]. Using a forward primer on intron 11 and a reverse primer on intron 10 [28], we obtained a PCR amplicon of about 2500 bp (Figure 4a). Sanger sequencing identified the proximal breakpoint of the duplication in an AluY element on intron 10 while the distal breakpoint was located in a highly homologous AluY on intron 11, with an overlapping sequence between the two introns of 47 base pairs (Figure 4b). The rearrangement resulted in a duplicated region of 5134 bp involving exon 11 and allowing for the accurate description of the duplication at the DNA level c.(3114-811_3202-1756)dup.

In the deleted samples, we applied a primer-walking strategy on intron 10 and intron 11. One of the primer pairs used was positioned near the 5′ end of exon 10 and the 3′ end of exon 11, respectively, resulting in a 2000 base pairs amplification product exclusively in the three deleted samples (Figure 5a). Sanger sequencing of the 2000 base pairs amplicons was used to delineate the deletion breakpoints (Figure 5b). The three patients carried the same deletion that can be described as c.(3114-5317_3202-3194)del at the DNA level. Both breakpoints were located in highly homologous AluSx repeat regions positioned in intron 10 and intron 11, and the rearrangement caused a deletion of about 8050 base pairs, leading to the complete loss of exon 11.

In Figure 6, the position of the deletion and duplication breakpoints with respect to the PALB2 domain is shown.

### 3.4. Haplotype Analysis

Haplotype analysis for chromosome 16 markers in the families clearly indicates that family 2 and 3 do not share the same haplotype on the deleted chromosome. In family 4, it was not possible to clearly define the affected haplotype, but some of the alleles were not compatible with the haplotype shared with the two other families (Figure 7).

## 4. Discussion

Our comprehensive genomic and transcriptomic analyses clarified the crucial role of *Alu* sequences to determine pathogenic large genomic rearrangements involving *PALB2* exon 11 in patients from Tuscany (Italy). In particular, we discovered a recurrent deletion [c.(3114-5317_3202-3194)del] in three different HBOC families and a duplication of the same exon in a fourth family [c.(3114-811_3202-1756)dup].

These out-of-frame deletions and duplications introduce a premature termination codon (PTC), which activates the nonsense-mediated decay (NMD) mechanism. NMD serves as a physiological surveillance pathway that facilitates the degradation of aberrant mRNAs, not only in the presence of PTCs. Mutations that trigger NMD are typically regarded as loss-of-function (LoF) events in the protein-coding genes they affect. However, genomic analyses have revealed that only a few PTCs can circumvent the NMD pathway, leading to the synthesis of frameshifted or truncated proteins [35]. Numerous studies underscore the critical role of NMD in relation to tumor suppressor genes (TSGs) and tumor development. The primary consequence of PTC mutations in TSGs appears to be an LoF effect due to NMD, which disrupts regulatory constraints on tumor growth, bypasses cell cycle controls, and confers a selective advantage to cancer cells [36,37].

In the present study, the breakpoints of the two large rearrangements within the *PALB2* gene were also characterized. Sequence analysis of the three families carrying the exon 11 deletion was performed and the probands shared identical breakpoint positions and sequences. Furthermore, the breakpoints of the duplication were the same as the one described by Bouras et al. in a patient from France [28].

There are two possible reasons why the breakpoints were identical in all families with the same rearrangement: it could be the result of either a founder mutation in the population of Tuscany or a recurrent rearrangement mediated by the same microhomology region within the similar pair of *Alu* repeats in each family.

In the deletion cases, the microsatellite analysis performed in the families suggested that they did not share the same haplotype associated with the rearrangement. The microsatellites utilized in our analysis are located approximately 2.7 Mb and 1.9 Mb on the 5′ and 3′ end of *PALB2,* respectively. Therefore, their distance does not exclude the possibility that a smaller haplotype is shared among the three families. However, while we found the deleted allele in 3 out of 1019 *PALB2* analyzed cases, in a consecutive series of 1019 individuals referred to our laboratory for BRCA analysis, we found 10 carriers of the oldest founder mutation discovered in Tuscany, the c.3228_3229delAG, with a mutation age estimated to be ~129 generations [38]. Indeed, we would expect a much higher frequency of families harboring the *PALB2* exon 11 deletion if a founder effect was present in Tuscany for this variant.

Germline rearrangements of *PALB2* represent a minor proportion of all genetic alterations associated with HBOC disorder; however, their identification provides important data for genetic counseling. The predilection for rearrangements involving exon 11 is intrinsically linked to the increased presence of *Alu* repeats within the region of the *PALB2* gene. Indeed, the *PALB2* gene lies on the short arm of human chromosome 16 and presents a high percentage of repeat elements. *Alu* repeats account for 64% of the *PALB2* intronic region, reaching 73% of the genomic sequence outside the exon–intron junctions. Essentially, these sequences offer many opportunities for homologous recombination, and nonallelic homologous recombination represents a common disease-causing mechanism associated with genome rearrangements [39]. We identified two Alu sequences located around the breakpoint junctions: (1) an *AluSx* within intron 10 and another one in intron 11 in the deleted patients and (2) *AluY* repeats in intron 10 and 11 of the *PALB2* gene, respectively, involved in the duplication. Therefore, our findings support the hypothesis that homologous recombination events underlie the identified rearrangements. Given its frequency, the deletion of exon 11 might represent a mutational “hotspot” of *PALB2* at least in the population of Tuscany; it might be particularly interesting to test the breakpoints in the other deletion cases described in the literature [29,32] to check if they involve the same pair of *AluSx* repeats. Likewise, the breakpoints identified in the exon 11 duplication were the same as those already reported in the literature [28,31], highlighting the possibility of another rearrangement hotspot involving *AluY*. Although all *Alu* repeats have an average of 85% homology [40], the two *AluY* repeats involved in the duplication are extremely homologous, with 92.17% identity, suggesting that recombination involving this pair may be particularly favorable.

In summary, this study highlights the noteworthy instability of intronic regions flanking exon 11 of *PALB2* and identifies a previously unreported hotspot involving *Alu* repeats with very high sequence homology in introns 10 and 11 of the gene.

## 5. Conclusions

In conclusion, this work showed that the method routinely adopted in our laboratory for the molecular diagnosis of hereditary breast–ovarian cancer led to the identification of recurrent rearrangements involving exon 11 of *PALB2* in Italian families, thus contributing to extending the mutational *PALB2* landscape in our population.

## Figures and Tables

**Figure 1 cancers-16-04022-f001:**
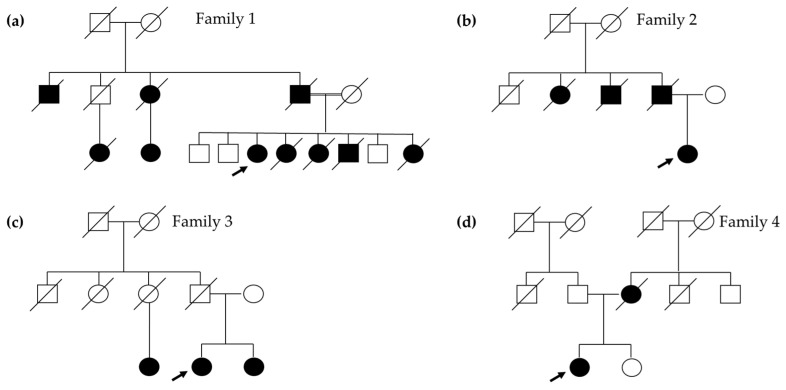
Pedigrees of individuals with *PALB2* rearrangements. (**a**) Family of Patient 1, carrier of exon 11 duplication. (**b**–**d**) Pedigrees of the three carriers of exon 11 deletion. The index case is indicated by an arrow.

**Figure 2 cancers-16-04022-f002:**
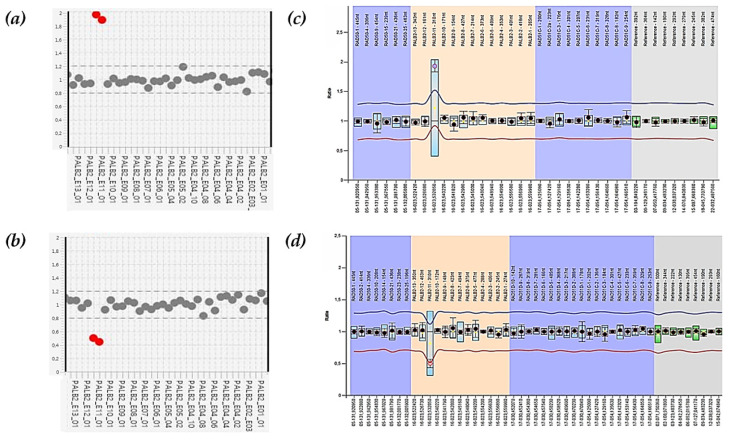
Comprehensive analysis of the *PALB2* gene. (**a**,**b**) CNV analysis by Amplicon Suite software v3.7.0 (SmartSeq srl, Novara, Italy) for woman carriers of duplication and deletion of *PALB2* exon 11. Plot of *PALB2* amplicons which show an increase of around 1.8–2 value of coverage in exon 11 (red dots) (**a**), and a reduction of about 0.5 value of coverage in exon 11 (red dots) (**b**); amplicon coverage of the other *PALB2* exons ranges to normal range (0.8–1.2), with a homogeneous distribution pattern of each amplicon. (**c**,**d**) MLPA assay of *PALB2* in carriers of exon 11 duplication (red dots) (**c**) and deletion (red dots) (**d**). Each dot represents the mean amplification ratio of specific region probes, reflecting the relative amount of the targeted DNA segment compared to a reference sample. Error bars show the standard deviation of the measured ratio. The ratio is calculated for each probe and indicates whether there is a deletion (ratio < 0.5), duplication (**c**) (ratio > 1.5) (**d**), or no change (ratio ≈ 1).

**Figure 3 cancers-16-04022-f003:**
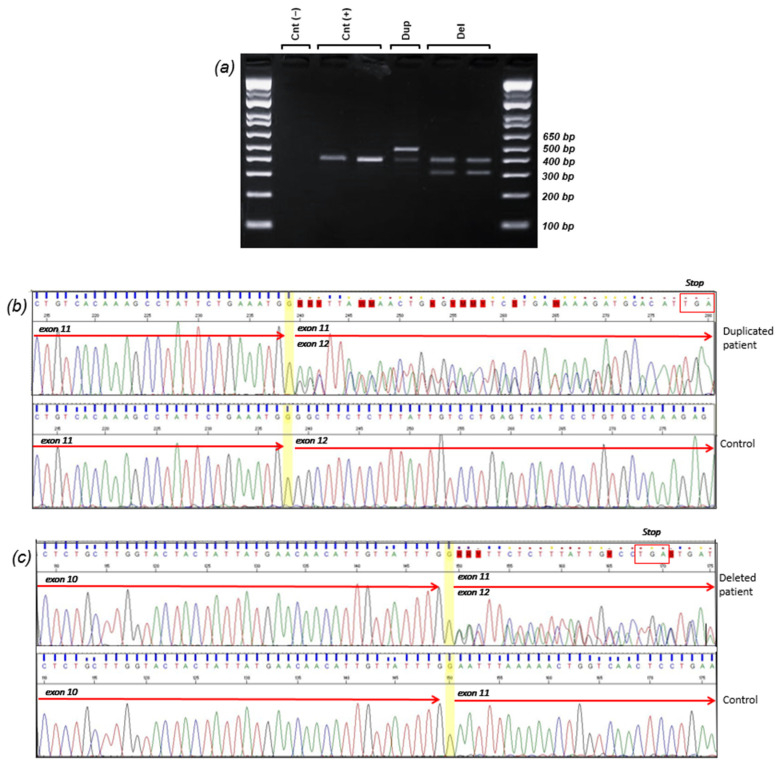
Large genomic rearrangements disrupt normal splicing, leading to frameshift mutations. (**a**) Agarose gel electrophoresis of reverse-transcription polymerase chain reaction (RT-PCR) conducted on mRNA isolated from patients and non-carrier controls. Cnt (−): No cDNA control; Cnt (+): wild-type band of 398 bp from healthy cDNA control; Dup: an additional band (486 bp) was observed in cDNA from Patient 1, who has an exon 11 duplication; Del: An extra band (310 bp) was detected in cDNA from Patients 2 and 3, both of whom have an exon 11 deletion. (**b**) Sanger sequencing of the two alleles from the RT-PCR product of Patient 1 (duplication): the insertion results in the formation of an early stop codon, as indicated in the red box. (**c**) Sanger sequencing of the two alleles from the RT-PCR products of the deleted patients: exon 11 deletion leads to a premature stop codon, as highlighted in the red box. The uncropped bolts are shown in Appendix A.

**Figure 4 cancers-16-04022-f004:**
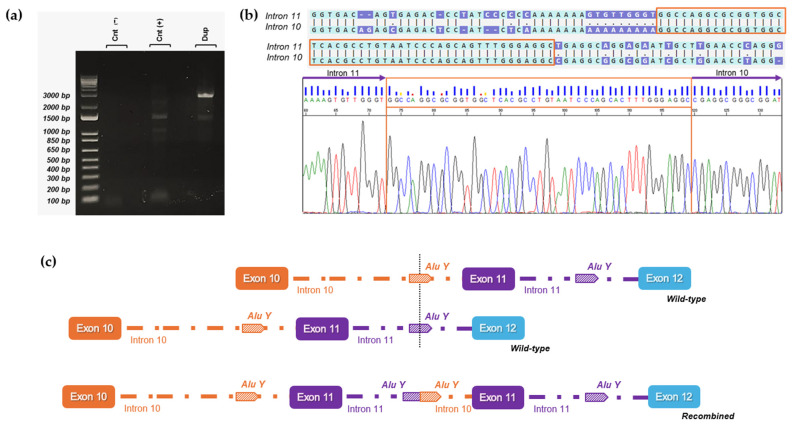
Breakpoint characterization of exon 11 PALB2 duplication. (**a**) Long-range PCR shows a specific band of approximately 2.5 kb detected only in the proband’s gDNA (Dup) sample and absent in control gDNA samples [Cnt(−), Cnt(+)]. (**b**) Electropherogram showing the breakpoint sequence (forward) where a tandem duplication site of 47 base pairs (TDS) is boxed in orange. (**c**) Schematic representation illustrates how wild-type alleles recombine in an unconventional manner via the AluY sequences located in introns 10 and 11, ultimately resulting in the formation of the deletion allele. The uncropped bolts are shown in Appendix A.

**Figure 5 cancers-16-04022-f005:**
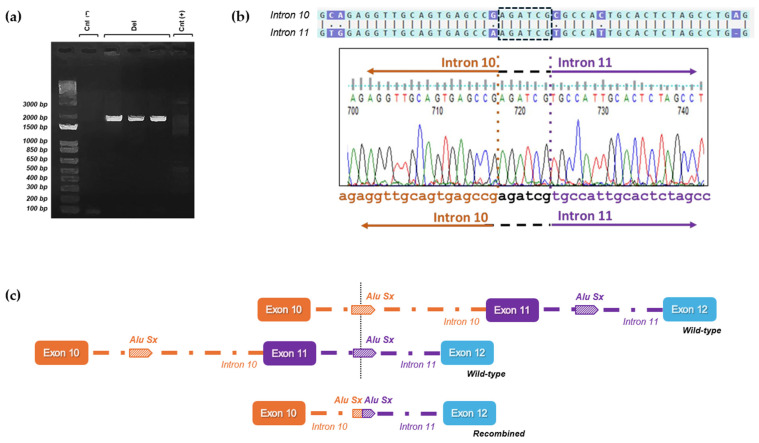
Breakpoint characterization of exon 11 PALB2 deletion. (**a**) Long-range PCR analysis revealed a 2 kb deletion product only in the gDNA deleted samples (Del), whereas healthy control samples exhibited no such product [Cnt(−), Cnt(+)]43. (**b**) Sanger sequencing of the 2 kb PCR product identifies the breakpoint regions of the rearrangement, characterized by a distinctive six-nucleotide element, AGATCG, that overlaps both breakpoints. (**c**) Schematic representation illustrates how wild-type alleles recombine in an unconventional manner via the AluSx sequences located in introns 10 and 11, ultimately resulting in the formation of the deletion allele. The uncropped bolts are shown in Appendix A.

**Figure 6 cancers-16-04022-f006:**
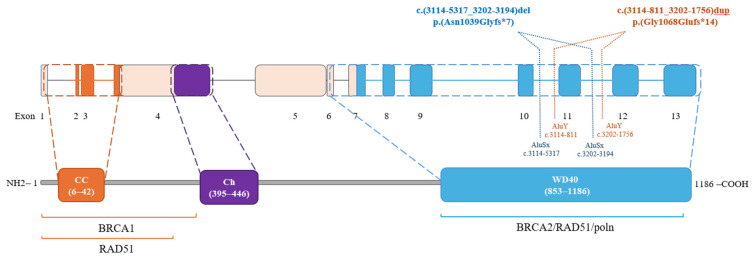
Schematic representation of PALB2 protein domains. CC: coiled coil motif (amino acids 6–42). Ch: chromatin-associated motif (amino acids 395–446), WD40: WD-like repeats (amino acids 853–1186). The binding regions with BRCA1 (amino acids 1–319), RAD51(amino acids 1–200; 853–1186), and BRCA2 (amino acids 853–1186) are also shown. In the WD40 domain, near exon 11, intronic breakpoints and the respective Alu repeats of the duplication and deletion are indicated.

**Figure 7 cancers-16-04022-f007:**
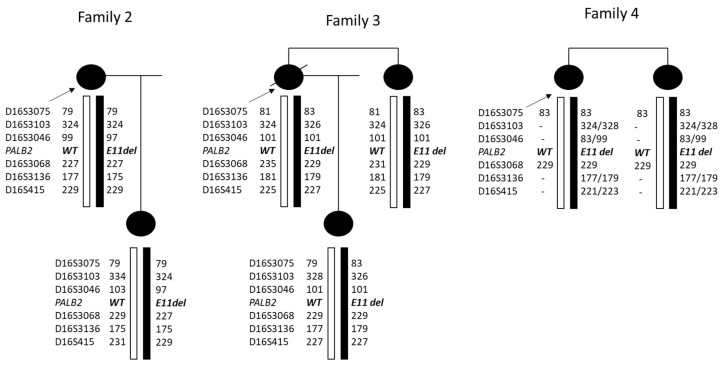
Haplotype analysis for chromosome 16 markers in the families clearly indicates that family 2 and 3 do not share the same haplotype on the deleted chromosome. In family 4, it was not possible to clearly define the affected haplotype, but some of the alleles were not compatible with a haplotype shared with the two other families.

**Table 1 cancers-16-04022-t001:** Position of STS markers and *PALB2* gene on chromosome 16 was established using the UCSC Genome Browser (URL (accessed on 10 September 2024): http://genome.ucsc.edu).

Sequence Map (hg19)	

12,209,198–12,209,445	D16S3075

17,473,463–17,473,796	D16S3103

20,886,398–20,886,710	D16S3046

23,614,627	**PALB2**
23,652,642	

25,560,601–25,560,948	D16S3068

50,706,233–50,706,482	D16S3136

53,670,661–53,671,041	D16S415

## Data Availability

The data presented in this study are available on request from the corresponding author due to privacy and ethical reasons.

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
