# Peer review of "Alu–Mediated Duplication and Deletion of Exon 11 Are Frequent Mechanisms of PALB2 Inactivation, Predisposing Individuals to Hereditary Breast–Ovarian Cancer Syndrome"

_cancers, 2024, doi:10.3390/cancers16234022_

Round 1
Reviewer 1 Report
Comments and Suggestions for Authors
Sidoti et al describe 4 families with inherited pathogenic rearrangements in PALB2. One is a duplication of exon 11 with the same genomic breakpoint as described in ref #28. The other three are a deletion of exon 11 with the same genomic breakpoints in each family. The three families do not share a microsatellite-defined haplotype suggesting they may not be related to each other although they are all from the same region of Italy. The genomic breakpoint information presented has not been published before as far as I am aware.
I suggest more discussion on the potential that the deletion mutation is a founder allele. The microsatellites used are located ~3Mb either side of PALB2 and in my opinion do not ‘exclude a founder effect’. It is possible that this in an ‘old’ mutation with a small (less than 6Mb) haplotype shared by the three families.
The material in the Introduction is very well established and could be shortened to one or two paragraphs. The description of the cancer features and age of diagnoses could be included in the pedigree figure, if allowed by the journal.
Author Response
Comment 1: I suggest more discussion on the potential that the deletion mutation is a founder allele. The microsatellites used are located ~3Mb either side of PALB2 and in my opinion do not ‘exclude a founder effect’. It is possible that this in an ‘old’ mutation with a small (less than 6Mb) haplotype shared by the three families.
Response 1: We thank the reviewer for this pertinent observation. We do agree that we cannot exclude an old founder mutation therefore we improved the discussion section with the following sentences (page 11, lines 334-343):
"The microsatellites utilized in our analysis are located approximately 2.7 Mb and 1.9 Mb on the 5’ and 3’ end of PALB2, respectively. Therefore, their distance does not exclude the possibility that a smaller haplotype is shared among the three families. However, while we found the deleted allele in 3 out of 1,019 PALB2 analyzed cases, in a consecutive series of 1,019 individuals referred to our laboratory for BRCA analysis, we found 10 carriers of the oldest founder mutation discovered in Tuscany, the c.3228_3229delAG, with a mutation age estimated to be ~129 generations [38]. Indeed, we would expect a much higher frequency of families harboring the PALB2 exon 11 deletion if a founder effect was present in Tuscany for this variant."
Comment 2: The description of the cancer features and age of diagnoses could be included in the pedigree figure, if allowed by the journal.
Response 2: We tried to include both cancer diagnosis and age of onset in the pedigrees but, unfortunately, the figure resulted too crowded and difficult to be read.
Reviewer 2 Report
Comments and Suggestions for Authors
The manuscript by Sidoti et al does not bring substantial novelties to the scenario currently known in the hereditary breast cancer syndrome. PALB2 is a gene that has long been routinely analyzed in patients affected by breast/ovarian cancer but without mutations in the BRCA1 and BRCA2 genes. However, as the authors of this manuscript state, the study contributes to broaden the mutational panorama of PALB2 in the analyzed population. Furthermore, the text is well written and the experimental procedures well described and adequate to achievement the result reported in the study.
Author Response
Response: Thanks very much for the positive comments, We do agree that our manuscript contributes to broaden the mutational panorama of PALB2.
Reviewer 3 Report
Comments and Suggestions for Authors
In this article, Sidoti et al focused on the mechanism leading to PALB2 exon 11 rearrangement found in patients with hereditary breast and ovarian cancer. They found AluY or AluSx elements to be associated with the observed exon 11 duplication or deletion, respectively. Both genomic events led to the production of truncated PALB2 proteins. While the result presented is solid, the scope of the study is a bit limited. I was hoping to see more insight on the biological consequences of these rearrangement events, and discussion on the clinical outcomes or importance, both of which are lacking from the article. The authors also did not show how such rearrangements lead to changes in the protein expression level and functions, which is important since the variants are heterozygous. In my opinion, more work needs to be done to show the importance of the results presented here. Apart from that, here are some minor suggestions:
1. Fig2: It would be great to include a schematic representing the PALB2 protein with protein domains, gene structures, and deletion/duplication positions.
2. Fig3/4: It would be great to have a PCR primer-template diagram to show the positions of the primers and expected size of amplicons etc.
3. Fig3: According to the PCR result, it seems that a higher number of transcripts are produced from the duplicated allele compared to the wild type allele, while similar number of transcripts are produced from the deletion allele compared to the wild type allele.
How does this affect the total amount of full-length vs. truncated protein produced? Any difference in the patient outcome? How is the patient outcome compared to other variants with exon 11 alteration?
Reviewer 4 Report
Comments and Suggestions for Authors
This study focused on large-scale gene rearrangements in exon 11 of the PALB2 gene, especially duplication and deletion variants, associated with the risk of familial breast-ovarian cancer syndrome (HBOC). The authors used next-generation sequencing (NGS) and multiplex ligation-dependent probe amplification (MLPA) to detect duplications and deletions in PALB2 exon 11; analyzed the effects of gene rearrangements in PALB2 exons on transcripts by reverse transcription-PCR (RT-PCR) and Sanger sequencing; and further precisely located the breakpoints of exon 11 variants by primer walking and long-fragment PCR. The study found that these gene rearrangements were mediated by Alu repeat sequences, resulting in the loss of function of the protein encoded by PALB2 due to premature termination. In addition, it was found that rearrangements caused by Alu repeat sequences were more frequent in the intronic region of the PALB2 gene, further revealing the instability of the exon 11 region of PALB2.
However, the small number of patient samples is a major drawback of this article. To enhance the statistical and population applicability of the results, increase the sample size of patients, especially those from diverse geographic and ethnic backgrounds; alternatively, consider explaining in more detail in the Discussion the impact of sample size limitations on the findings.
Author Response
Comment 1: The small number of patient samples is a major drawback of this article. To enhance the statistical and population applicability of the results, increase the sample size of patients, especially those from diverse geographic and ethnic backgrounds; alternatively, consider explaining in more detail in the Discussion the impact of sample size limitations on the findings.
Response 1: We do agree that a larger sample might permit more definite conclusions. However, this manuscript documents, for the first time, the breakpoints of deletion involving PALB2 exon 11. A larger sample will help ascertain the frequency of such deletions and duplications across different populations, patterns that we believe to be quite rare.
Round 2
Reviewer 3 Report
Comments and Suggestions for Authors
The authors have adequately addressed my comments.